# Post-Diagnostic Support for Behaviour Changes in Young-Onset Dementia in Australia

**DOI:** 10.3390/brainsci13111529

**Published:** 2023-10-30

**Authors:** Claire J. Cadwallader, Dennis Velakoulis, Samantha M. Loi

**Affiliations:** 1Neuropsychiatry, Royal Melbourne Hospital, Parkville, VIC 3050, Australia; cjcadwallader94@gmail.com (C.J.C.); dennis.velakoulis@mh.org.au (D.V.); 2The Turner Institute for Brain and Mental Health, Monash University, Clayton, VIC 3168, Australia; 3Department of Psychiatry, University of Melbourne, Parkville, VIC 3052, Australia

**Keywords:** young-onset dementia, behaviour changes, formal support services

## Abstract

Behaviour changes (BCs) are common in young-onset dementia (YOD). Access to knowledgeable and age-appropriate support services is needed to assist with the appropriate management of BCs. We sought to investigate the types of YOD-related BCs that most commonly require support, the formal services being accessed for support, and the experiences of those seeking support in Australia. We employed a cross-sectional online questionnaire for individuals living with YOD as well as individuals providing informal or formal care for someone with YOD. Thirty-six questionnaire responses were recorded. Of the total sample, 83% reported YOD-related BCs requiring support, the most common being appetite/eating changes, followed by agitation/aggression and apathy/indifference. Seventy-six percent of these individuals had attempted to seek support from a formal service, with Dementia Australia, Dementia Support Australia, and general practitioners most commonly approached. Responses suggested that the support access pathway is suboptimal, with a lack of clarity about what services to approach for support and long wait times. Furthermore, 28% of participants had not gained access to support utilizing non-pharmacological strategies. Individuals who need support for YOD-related BCs demonstrated a high rate of help-seeking from formal support services; however, the support access pathway is slow, is difficult to navigate, and does not result in the best-practice management of BCs. Formal services resourced to provide efficient support with the implementation of behavioural strategies are needed, along with clear, accessible guidelines on the pathway to access them.

## 1. Introduction

Young-onset dementia (YOD) is characterised by symptom onset prior to 65 years of age and accounts for 5–10% of diagnosed dementias in Australia [1,2,3,4]. Non-cognitive symptoms or behaviour changes (BCs) associated with YOD include lowered mood, anxiety, agitation, psychotic symptoms, apathy, and sleep and appetite changes [5]. BCs are more prevalent and severe in YOD compared to older-onset dementia (OOD) [6,7,8,9]. These changes result from organic pathology associated with YOD and in response to unmet needs of the individual (e.g., unmanaged pain, thirst, and psychological distress) [10]. BCs in YOD have a significant negative impact on the wellbeing of the individual and their carers. For the individual, these changes contribute to poor quality of life [11] and predict earlier entry into residential care [12]. Compared to caregivers of people with OOD, caregivers of people with YOD experience greater perceived difficulties due to BCs [13], as well as more distress, anger, depression, and feelings of being overwhelmed [7,9].

Behavioural strategies are effective for managing BCs and are recommended as first-line strategies or to be used in combination with pharmacological interventions [14,15]. Easy access to knowledgeable YOD-appropriate support services that can facilitate the implementation of these strategies is lacking. Furthermore, there are no current guidelines on how and where to seek support for BCs in YOD in Australia. Dementia support services are typically designed for older adults and are less effective at addressing the unique psychosocial challenges experienced by those with YOD [16]. Those with YOD fall through the gaps of the funding and eligibility structures surrounding these support services, which are often focused on older adults [16]. An Australian YOD-specific ‘key worker’ programme, under the auspices of Dementia Australia, was established in 2013 to help those affected by YOD navigate available supports [17]. This program was discontinued with the introduction of the National Disability Insurance Scheme (NDIS) scheme in 2019 for which people with YOD were eligible. Although access to support through the NDIS has generally been received favourably [18], those seeking support for the management of BCs in YOD have more difficulty accessing appropriate services without knowledgeable YOD-specific case management.

O’Connor and colleagues [19] investigated the experiences of family caregivers in managing BCs in YOD in an Australian sample. BCs that were difficult to manage were reported by over 90% of caregivers, with variable confidence rated in their ability to manage these and in accessing formal supports. It was concluded that better access to YOD-specific support services for behaviour change management was needed to improve confidence and competence in caregivers. This study did not specifically seek information regarding the BCs for which support was being sought (as opposed to BCs that were present), different formal support services that had been accessed specifically for support with BCs, or people’s experiences with these. In order to gain a full picture of the service gaps and requirements, it is important to capture the experiences and needs of people with YOD themselves as well as those providing care to people with YOD such as healthcare professionals and paid caregivers. The aim of the current study was to examine the specific BCs for which individuals with YOD and those caring were seeking support, the types of formal support being sought for these BCs, and the satisfaction with this support, in an Australian sample of people with YOD and their caregivers.

## 2. Materials and Methods

A cross-sectional online questionnaire was developed for people living with YOD (PLWYODs) or individuals providing informal or formal care, including clinicians and service providers, for someone with YOD. Questionnaire responses were collected between 12 December 2022 and 4 April 2023. Participants were recruited via posts on social media, flyers circulated to specialist YOD clinics, and snowball sampling. Eligibility criteria required that participants were either (a) a PLWYOD (onset of symptoms prior to 65 years of age), (b) an informal caregiver (unpaid) living with someone who has YOD, or (c) a formal caregiver (e.g., healthcare practitioner or paid caregiver) providing care to at least one person with YOD. All participants signed an online consent form. This study was ethically approved by the University of Melbourne Central Human Research Ethics Committee (CHREC), project ID #23580.

The questionnaire (Appendix A) was conducted through the RedCAP platform, and was aimed at gathering information about the types of BCs people with YOD were presenting with and participants’ experiences with seeking support for the management of BCs. General demographic information was collected. PLWYODs and informal caregivers were also asked to provide information about the diagnosis of YOD, including dementia type, the age of onset, and the type of clinician who provided the diagnosis. These questions were not asked of formal caregivers, as they were likely to have cared for more than one person with YOD.

Informal caregivers completed the Neuropsychiatric Inventory Questionnaire (NPI-Q), an informant questionnaire assessing symptoms present such as agitation, mood changes and sleep, their severity and associated caregiver distress [20]. Each symptom domain is rated on the presence, severity (“mild” to “severe”), and the associated distress (“not at all” to “very severely”), with higher scores indicating worse severity and distress.

All participants were asked if they had noticed BCs related to the dementia that they required advice or support for, and whether they had previously sought formal support for these. The questionnaire was discontinued if a participant answered ‘no’ to either of these questions. The remaining items in the questionnaire related to participants’ experiences with seeking and receiving support for BCs. These included which formal services participants had attempted to seek support from, how long it took to receive support, and whether they were given a recommendation of medication or non-pharmacological strategies. Participants were asked to rate their agreement with statements related to satisfaction with the process of accessing support, as well as the support received.

The questionnaire was piloted by several people with lived experience who provided feedback that influenced the final design, including the language used, the scope of the questions, and the time it took to complete. This included two PLWYODs, two informal caregivers of PLWYODs, and four formal caregivers (three clinicians and one formal caregiver).

The secure platform RedCAP was used to collect and store the data, and to generate descriptive statistics (i.e., percentages, means, and standard deviations).

## 3. Results

Thirty-six people completed the questionnaire. Participants were 4 PLWYODs, 17 informal caregivers for someone with YOD, and 15 formal caregivers for someone with YOD. Participant characteristics are presented in Table 1.

PLWYODs were on average 58 years old (SD = 7.27, 2 (50%) females), all of whom were living in the community. Informal caregivers were on average 59 years old (SD = 15.77; 14 (82%) females), and the majority were supporting a person with YOD living in the community (11/17). Formal caregivers were on average 46 years of age (SD = 12.14; 6 (40%) female; 1 (6%) preferred not to identify gender). The majority of participants (27 (75%)) lived in Victoria, while 4 (11%) lived in South Australia, 3 (8.5%) lived in New South Wales, and 2 (5.5%) lived in Queensland.

### 3.1. Details of YOD

*PLWYOD* reported a mean age of symptom onset of 47 years old (SD = 11.79). One was diagnosed with Alzheimer’s disease (AD), two were diagnosed with language-variant frontotemporal dementia, and one had dementia with Lewy bodies. *Informal caregivers* reported a mean age of symptom onset of 55 years old (SD = 6.73) and a mean age at diagnosis of 57 years old (SD = 7.09). Diagnoses of the person with YOD were mostly AD (n = 8) and fronto-temporal dementia (FTD) (n = 5). The majority of dementia diagnoses were given by neurologists. Informal caregivers completed the NPI-Q. The mean total symptom score was 4.33 (SD = 2.55), the mean severity score for reported BCs was 9.86 (SD = 5.88), and the average caregiver distress score was 11.71 (SD = 7.73).

### 3.2. Behaviour Changes

A flow chart of reported BCs and the details of the support sought and received for BCs are presented in Figure 1. Eighty-three percent of the sample reported at least one BCs for which they needed support. The most common types of BCs for which participants needed support were appetite/eating changes (63%), agitation/aggression (60%), and apathy/indifference (60%). The majority (76%) of participants who reported BCs had attempted to seek management support from at least one formal service. The most common services approached for support were Dementia Australia (48%), Dementia Support Australia (48%), and general practitioners (GPs) (48%). Five (22%) and two (9%) respondents nominated adult and aged mental health services as a source of support for BCs, respectively. Ninety-one percent had received some form of support with a most common wait time of within three (24%) and six (29%) months. The most common modes of support people received were over the phone (50%), home visit assessments (30%), and in-person appointments (25%). Most people were recommended both pharmacological and behavioural strategies to manage the BCs (43%), while some received only medication recommendation (28.5%) and others were recommended only behavioural strategies (28.5%).

Key results for BCs requiring support, formal services approached for support, and recommendations received for the management of BCs are presented for each group separately (i.e., PLWYODs, informal caregivers, and formal caregivers) in Figure 2. While the most common BCs nominated by PLWYODs and informal caregivers was eating/appetite changes (100% and 73%, respectively), agitation/aggression was the most common BC requiring support for formal caregivers (81%) (Figure 2A). PLWYODs and informal caregivers were more likely to approach GPs (75% and 50%, respectively) and Dementia Australia (75% and 42%, respectively), while formal caregivers sought support from Dementia Services Australia (63%) and adult mental health services (38%) most often (Figure 2B). Finally, recommendations given to the three groups were mixed (Figure 2C). PLWYODs and informal caregivers were, respectively, recommended medication only 0% and 36% of the time, behavioural strategies only 67% and 36% of the time, and a combination 33% and 18% of the time. Formal caregivers were most likely to have been recommended a combination of behavioural strategies and medication (75%), while 25% were recommended medication only.

### 3.3. Satisfaction with Services

Satisfaction ratings for the ease of accessing support services for BCs in YOD are presented in Figure 3. Three informal caregivers and five formal caregivers did not complete this section of the survey, leaving 28 respondents. In general, most (72%) PLWYODs and informal caregivers and some (30%) formal caregivers “completely agreed” or “agreed” that they did not know where to start when looking for a service to help them manage BCs. Most (71%) PLWYODs and informal caregivers and some (40%) formal caregivers “agreed” or “completely agreed” that the process of accessing services to help with BCs was difficult. Most participants in all groups “completely agreed” or “agreed” that accessing services took too long (83% of PLWYODs and informal caregivers; 60% of formal caregivers). Finally, the majority of participants “completely agreed” or “agreed” that having guidelines for who to contact when support for BCs was needed would be helpful (89% of PLWYODs and informal caregivers; 90% of formal caregivers). Satisfaction ratings for individual support services are presented in Appendix A. Due to low response rates for these questions, reliable comparisons between services were not possible.

## 4. Discussion

In this study, we examined the experiences of PLWYODs and those that provide care for them in seeking and receiving support for the management of BCs in YOD in Australia. The most frequent YOD diagnoses were of AD and FTD. Of a total of 36 respondents, over 80% reported BCs for which they required support, the most common being appetite/eating changes, followed by agitation/aggression and apathy/indifference. Over 75% of these participants had attempted to seek support from a formal service, with Dementia Australia, Dementia Support Australia, and GPs most commonly approached for support. There were differences in the BCs nominated and type of services contacted, depending on whether the respondents were formal or informal caregivers or PLWYODs. Wait times for support were between three to six months. Just over two-thirds of the participants had been recommended behavioural strategies, either in combination with pharmacological intervention or as the stand-alone management option. Of concern, almost a third of participants were recommended pharmacological intervention only. Finally, most participants agreed that wait times to receive support from services were too long and that guidelines were needed on who to contact when support for BCs is needed. The majority of PLWYODs and informal caregivers, and some formal caregivers, agreed that it was difficult to know where to start when seeking access to a support service for management of BCs and that accessing an effective support service was difficult.

Participants endorsed an average of four BCs that required support. Interestingly, our study highlighted a difference in the most common YOD-related BCs individuals report they need support for, compared to the most common BCs reported as present in other studies. Consistent with previous studies reporting on specific BCs with a high prevalence in YOD [6,8,21], we found that agitation/aggression and apathy/indifference were among the most common BCs for which people required management support. However, the most common BCs for which participants required support were eating/appetite changes (over 60% of the sample). This appeared to be driven mostly by responses from PLWYODs and informal caregivers. Previous studies have shown that eating changes are prevalent in YOD [6,21]. Our results demonstrate that eating changes are an important area for which PLWYODs and caregivers require support, and clinicians should include such enquiries in their assessments. Eating changes in dementia can be due to alterations in appetite, cognition, and/or diet [22,23], e.g., malnutrition, food refusal, slow eating, rigidity in food choices, and hyperphagia. There is a sparse literature on interventions to address eating changes in individuals with dementia; however, non-pharmacological strategies, such as monitoring and supervising meals and changing food texture and appearance, may be effective [24,25]. Formal caregivers most often required support for agitation/aggression and apathy/indifference, which may reflect the high prevalence rate of this type of BC presenting in the clinical setting [6,8,21].

In contrast with previous studies that have shown low access rates for formal support services by PLWYODs and caregivers [16,26,27], over 75% of respondents who reported needing management support for YOD-related BCs in the current study had attempted to access a formal support service. Among the commonly accessed services across all participants were Dementia Australia and Dementia Support Australia. This is important in that those Australian Commonwealth agencies that are advertised as providing support and strategies for BCs are being appropriately accessed as such. GPs were also a common service used by PLWYODs and informal caregivers for support of BCs. Although our survey could not identify the specific type of interventions GPs provided to participants, a recent systematic review [28] and other related research [29] suggested that GPs were more likely to prescribe medication for treatment of BCs and felt “less comfortable” with providing non-pharmacological strategies.

While our results show a high rate of help-seeking from formal services appropriate for the management of BCs in dementia, participants’ experiences of seeking this support were rated less favourably. Participants, particularly PLWYODs and informal caregivers, had difficulty knowing where to start when looking for a service for support, and generally found the process of accessing support difficult. Furthermore, a majority of participants indicated that the process of receiving support took too long. Our results highlight that BCs requiring support are common in YOD and that the access pathway for support of YOD-related BCs needs improvement in clarity and efficiency. In Australia, public mental health services, whether such services are for those aged more than 65 years old or for adults, may be utilised by individuals looking for support for BCs in YOD. We found that, while these services were more often accessed by formal caregivers, they were accessed less by PLWYODs and informal caregivers, suggesting a lack of awareness that these services exist and provide this support. A better integrated model of care whereby YOD specialist services are embedded within either adult or aged mental health services has been recommended [30] and investigating the use of mental health services by individuals with YOD will provide an improved understanding of current service utilisation. Furthermore, general practitioners, who were commonly approached for support in our sample, could be educated about specialised services that provide support for BCs in YOD so that appropriate referrals are made.

In addition to an examination of the experiences of people seeking help for BCs in YOD, the current study sought to understand the type of support and recommendations people received from formal support services. BCs in dementia are often the result of the unmet needs of the individual [14]. Behavioural strategies that address these underlying causes have become the gold-standard recommendation [15], but few studies examining these have been published. For example, positive behaviour support programs have shown efficacy for addressing BCs such as apathy and disinhibition in FTD [31,32]. In our study, two-thirds of the participants had been recommended behavioural strategies, either alone or in combination with psychotropic mediation, while almost a third of the participants had only been recommended medications. Given the limited empirical evidence for psychotropic mediations as a stand-alone treatment in the management of BCs in dementia, along with the risk of unwanted side effects [33], this is an area of concern. The implementation of effective behavioural strategies requires an individualised approach that is more complex and time-consuming than the prescription of medication [15]. This highlights the need for well-resourced, knowledgeable, formal services that can support the implementation of behavioural strategies for YOD, ensuring that PLWYODs and caregivers are included in discussions about these management options to educate and inform their needs.

This study was limited by a small sample size, with participant residence predominantly in the state of Victoria, Australia. While there was equal representation from informal and formal care providers, the sample included only four people living with YOD. Our results may be less representative of the views of PLWYODs who need support for BCs. It is also possible that participants who chose to complete the questionnaire were more likely to have been dissatisfied with the services they received in the past. Due the low response rate, a more detailed investigation into the participants’ satisfaction with the support received from specific formal support services was not possible. These questions could be addressed in future studies with access to larger recruitment pools of PLWYODs and caregivers.

## 5. Conclusions

This study examined the experiences of PLWYODs as well as informal and formal caregivers in seeking support for the management of BCs in YOD in Australia. The most common BCs requiring support was eating/appetite changes for PLWYODs and informal caregivers, and agitation/aggression for formal care providers. In general, the three formal services utilised by participants were the two major governmental dementia agencies, Dementia Australia and Dementia Support Australia, and GPs. However, the pathway for seeking formal support and obtaining appropriate management for these BCs was suboptimal. We recommend the creation of clear, accessible guidelines on how and where to seek support for BCs in YOD in Australia. These results also highlight the importance of knowledgeable support services that can provide timely help and education for the implementation of behavioural strategies for BCs.

## Figures and Tables

**Figure 1 brainsci-13-01529-f001:**
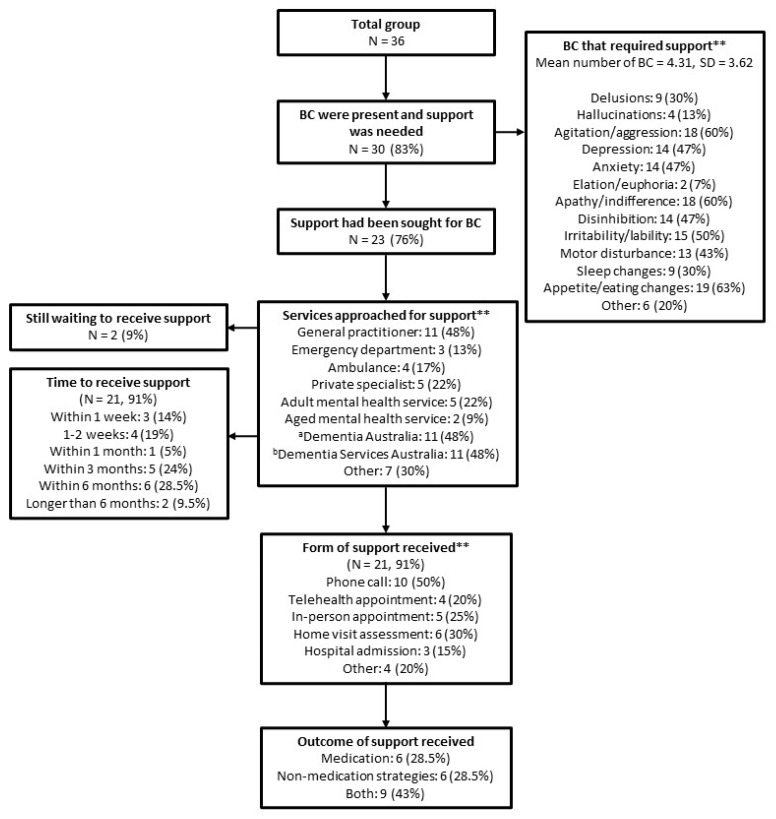
Flow diagram for reported behaviour changes and details of services approached for support. BCs = Behaviour changes. ** Participants selected multiple responses where applicable. Dementia Australia (www.dementia.org.au) (accessed on 28 September 2023) is a national not-for-profit organisation that provides education and support for those living with dementia and people who care for them. This includes a “Dementia Helpline” where support for BCs, among other challenges faced by PLWYODs and caregivers can be received over the phone. Dementia Support Australia (www.dementia.com.au) (accessed on 28 September 2023) is a government funded organisation specifically for management support of BCs in dementia, with a focus on behavioural strategies. They provide support via phone, video conferencing, or face-to-face meetings, and include a Dementia Behaviour Management Advisory Service (DBMAS) and a Severe Behaviour Response Team (SBRT).

**Figure 2 brainsci-13-01529-f002:**
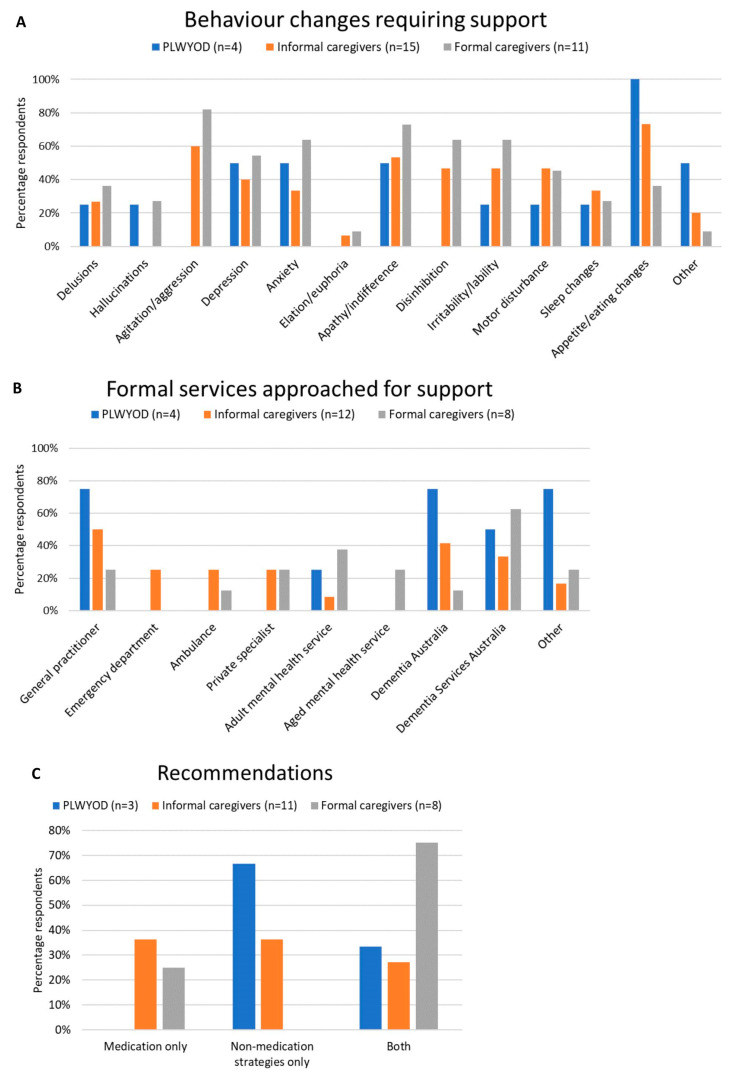
Behaviour changes requiring support (**A**), formal services approached for support (**B**), and recommendations received for the management of the behaviour changes (**C**), for PLWYODs, informal caregivers, and formal caregivers separately. Note that participants are not included in the above graphs if they responded that they had not noticed any BCs for which they needed support (**A**–**C**), or if they responded that they had not attempted to seek support from a formal service (**B**,**C**). One PLWYOD and one informal caregiver had approached a formal service but were still waiting to receive support, and were therefore, not included in (**C**).

**Figure 3 brainsci-13-01529-f003:**
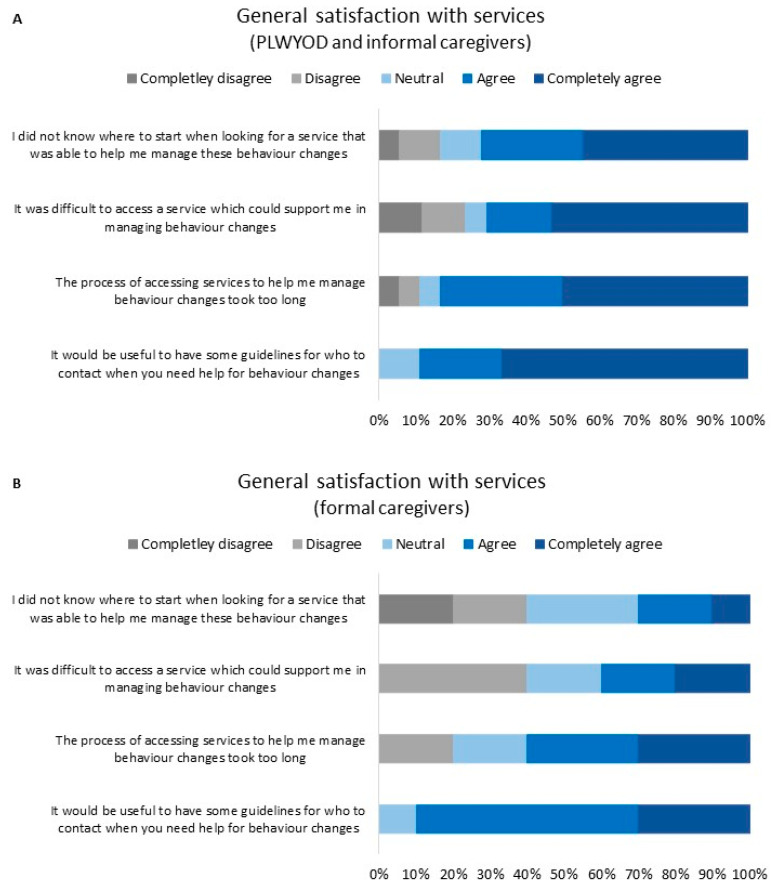
General satisfaction ratings with regard to formal support service access for YOD-related BCs. Responses from PLWYODs and informal caregivers (N = 18) (**A**); responses from formal caregivers (N = 10) (**B**).

**Table 1 brainsci-13-01529-t001:** Characteristics of the sample.

	Person with YOD n = 4	Informal Caregiver n = 17	Formal Caregiver n = 15	Total n = 36
Mean age, years (SD, range)	58.25 (7.27, 51–65)	59.59 (15.77, 32–80)	46.07 (12.14, 28–65)	53.80 (14.88, 28–80)
Gender				
Female	3	14	6	23
Male	1	3	8	12
Prefer not to say	0	0	1	1
Living situation of person with YOD				
In community with partner/family	4	11	-	15
Residential care facility	0	4	-	4
Supported independent living	0	1	-	1
Unsure	0	1	-	1
Formal caregiver type				
Medical	-	-	9	-
Nursing	-	-	2	-
Allied health ^1^	-	-	4	-
Rurality of residence				
Metropolitan/urban	2	12	-	14
Rural	2	5	-	7
Dementia type ^2^				
Alzheimer’s disease	1	8	-	9
Frontotemporal dementia	0	5	-	5
Language variant dementia	2	1	-	3
Behavioural variant dementia	0	3	-	3
Vascular dementia	0	2	-	2
Alcohol-related dementia	0	2	-	2
Dementia with Lewy bodies	1	0	-	1
Mean age, years, at symptom onset (SD, range)	47.3 (11.8, 35–63)	55.6 (6.7, 43–62)	-	53.9 (8.33, 35–63)
Mean age, years, at diagnosis (SD, range)	53.7 (10.0, 46–65)	57.8 (7.09, 47–65)	-	57.2 (7.4, 46–65)
Clinician who delivered diagnosis				
Psychiatrist	1	0	-	1
Neurologist	1	7	-	8
Geriatrician	1	2	-	3
Neuropsychologist	0	5	-	5
Allied health professional	1	0	-	1
Other	0	2	-	2
Unsure	0	1	-	1
Neuropsychiatric Inventory Questionnaire (NPI-Q)				
Mean total symptoms (SD, range)	-	4.33 (2.55, 1–9)	-	-
Mean severity (SD, range)	-	9.86 (5.88, 1–19)	-	-
Mean caregiver distress (SD, range)	-	11.71 (7.73, 2–28)	-	-

^1^ Allied health in Australia include professions such as occupational therapy, physiotherapy, psychology, social work, speech pathology, exercise physiology, music therapy etc. For more information see https://ahpa.com.au/allied-health-professions/ (accessed on 25 October 2023). ^2^ Some informal caregivers reported more than one diagnosis.

## Data Availability

The data are not publicly available due to privacy considerations surrounding the inclusion of participants living with, or caring for someone living with, rare forms of young-onset dementia.

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
