# Peer review of "Post-Diagnostic Support for Behaviour Changes in Young-Onset Dementia in Australia"

_brainsci, 2023, doi:10.3390/brainsci13111529_

Round 1

Reviewer 1 Report

Comments and Suggestions for Authors

This is a fine paper. Motivation, aim, methods and results are clearly described. The overall merit is somewhat limited, though, due to the small number of respondents. I have only a few remarks on details which could improve the paper.

1) The sample is a convenience sample, and largely self-selected. It is possible that mainly people have responded who were not satisfied with the services received. This possible source of bias should be mentioned among the limitations of the study.

2) What kind of professionals were the formal caregivers in the sample? Since they also seem badly informed, it would be useful to have some information about this.

3) It would be useful to break Figure 3 down by type of respondent, or at least to distinguish between formal caregivers and others. One would expect the former to have fewer problems finding an appropriate service.

4) Possibly, a promising way to make it easier to identify and access suitable services seems to be to inform general practicioners about the options. Figure 1 shows that the general practicioner is the service most often approached. If this makes sense, could you comment on this proposal in the paper?

Comments on the Quality of English Language

The English is good, except there are two lapses:

- lines 162-164. Sentence seems ungrammatical

- line 212 One "that" too many.

Author Response

We thank the reviewers for their thoughtful feedback and useful suggestions on our manuscript titled “Post-diagnostic support for behaviour changes in young-onset dementia in Australia”. This input has been carefully considered and used to improve the quality of the manuscript. Please find our point-by-point response to each of the reviewer’s comments below.

Reviewer 1

  • The sample is a convenience sample, and largely self-selected. It is possible that mainly people have responded who were not satisfied with the services received. This possible source of bias should be mentioned among the limitations of the study.

We acknowledge that this limits generalisation of our findings.  In the limitations section of the study, we have added the sentence “It is also possible that participants who chose to complete the questionnaire were more likely to have been dissatisfied with the services they received in the past” (lines 299-300)

  • What kind of professionals were the formal caregivers in the sample? Since they also seem badly informed, it would be useful to have some information about this.

    We agree with this useful comment. We have added a section in Table 1 called ‘Formal caregiver type’ which distinguishes whether formal caregivers were in a medical, nursing or allied health profession. Most of the formal caregivers were in a medical profession (n= 9), while 4 were in an allied health profession, and 2 were in a nursing profession. A definition of allied health professions in Australia has been given in the Table footer.

  • It would be useful to break Figure 3 down by type of respondent, or at least to distinguish between formal caregivers and others. One would expect the former to have fewer problems finding an appropriate service.

    As suggested we have split the graph in Figure 3 into two graphs; one for PLWYOD/informal caregivers, and one for formal caregivers. We have also updated the reporting in the results section (lines 171-181), and the discussion (lines 222-227, and 261) to reflect these separated results.

  • Possibly, a promising way to make it easier to identify and access suitable services seems to be to inform general practitioners about the options. Figure 1 shows that the general practitioner service is most often approached. If this makes sense, could you comment on this proposal in the paper?

    Thank you for this suggestion. We have added the sentence “Furthermore, general practitioners, who were commonly approached for support in our sample, could be educated about specialised services that provide support for BC in YOD so that appropriate referrals are made.” (lines 274-277)

  • Lines 162-164. Sentence seems ungrammatical.

    Thank you. We have corrected this sentence to “Formal caregivers were most likely to have been recommendation a combination of behavioural strategies and medication (75%), while 25% were recommended medication only”

  • Line 212. One “that” too many

    The extra “that” has been deleted.

Reviewer 2 Report

Comments and Suggestions for Authors

Post-diagnosis care for people with dementia in Australia has been criticised for being insufficiently available, fractured, and focused on managing impairments rather than promoting wellbeing. Rates of dementia diagnosis remain low, and services available after diagnosis will depend on demographic and practitioner factors. Behaviour hanges. These problems are common to other high-income countries internationally, though they are exacerbated by Australia’s very low population density. Behaviour changes (BC) are common in young-onset dementia (YOD). Access to knowledgeable and age-appropriate support services is needed to assist with the appropriate management of BC. The authors sought to investigate the types of YOD-related BC that most commonly require support and employed a cross-sectional online questionnaire for individuals living with YOD and individuals providing informal or formal care for someone with YOD. Auhor’s has recommended the creation of clear, accessible guidelines on how and where to seek support for BC in YOD in Australia. These results also highlight the importance of knowledge-based support services that can provide timely help and education for the implementation of behavioural strategies for BC. The paper is generally well written and structured, but in my opinion, it has some shortcomings. My suggestions are as follows:

  1. A few effective strategies should be highlighted to bridge the ‘evidence-practice gap’ in dementia care and attract more citations and readers.
  2. How these data are taken is to improve adherence to the key recommendations from the Australian guidelines.
  3. How these data are taken is to People with dementia living in the community should be educated about the implementation of behavioural strategies for BC. reflecting evidence-based programmes.
Comments on the Quality of English Language

Authors should ensure that their article has been carefully checked for spelling, language, grammar, and style (where appropriate)

Author Response

We thank the reviewers for their thoughtful feedback and useful suggestions on our manuscript titled “Post-diagnostic support for behaviour changes in young-onset dementia in Australia”. This input has been carefully considered and used to improve the quality of the manuscript. Please find our point-by-point response the reviewer’s comments below.

Reviewer two

  • A few effective strategies should be highlighted to bridge the ‘evidence-practice gap’ in dementia care and attract more citations and readers.

We agree that effective non-pharmacological strategies are needed for the management of behaviour change in YOD and while the purpose of this study was not investigating the effectiveness of these for BC, we have included that there is very little published in the area of non-pharmacological interventions for BC in YOD (lines 282-283) and also positive behaviour support programs for frontotemporal dementia (lines 283-284).

  • How these data are taken is to improve adherence to the key recommendations from the Australian guidelines.

There are currently no Australian guidelines on seeking support for behaviour changes in YOD. We have made this clearer by including a sentence in the introduction that states “Furthermore, there are no current guidelines on how and where to seek support for BC in YOD in Australia”. This study is the first to investigate formal service use and satisfaction of services that offer support for BC in YOD in Australia. Our study emphasises and recommends “the creation of clear, accessible guidelines on how and where to seek support for BC in YOD in Australia” (lines 44-45).

  • How these data are taken is to people with dementia living in the community should be educated about the implementation of behavioural strategies for BC. Reflecting evidence-based programmes.

The data reported in this study does not directly inform how people living with dementia should be educated about behavioural strategies for BC. However, we agree that education about these strategies is important and that people living with YOD and their caregivers should be included in conversations about behavioural strategies and why these are effective. We have added a sentence to the discussion that states “This highlights the need for well-resourced, knowledgeable, formal services that can support the implementation of behavioural strategies for YOD, ensuring that PLWYOD and caregivers are included in discussions about these management options to educate and inform their needs.” (lines 293-294)